# Vegetation Height as the Primary Driver of Functional Changes in Orthopteran Assemblages in a Roadside Habitat

**DOI:** 10.3390/insects13070572

**Published:** 2022-06-24

**Authors:** Fran Rebrina, Klaus Reinhold, Nikola Tvrtković, Vesna Gulin, Andreja Brigić

**Affiliations:** 1Department of Biology, Faculty of Science, University of Zagreb, 10000 Zagreb, Croatia; vesna.gulin@biol.pmf.hr (V.G.); andreja.brigic@biol.pmf.hr (A.B.); 2Evolutionary Biology, Faculty of Biology, Bielefeld University, 33615 Bielefeld, Germany; klaus.reinhold@uni-bielefeld.de; 3Alagovićeva ulica 21, 10000 Zagreb, Croatia; nikolatvrtkovic71@gmail.com

**Keywords:** anthropogenic disturbance, grasshopper, cricket, spatial dynamics, functional diversity, environmental factor, road ecology

## Abstract

**Simple Summary:**

This paper reports the results of a field research that investigates functional changes in grasshopper and cricket (Orthoptera) assemblages with distance from a major road (at 10, 25, 50, 100 and 500 m). Expanding on a previous study that adopted a species-based approach to the same subject, in order to gain a different perspective on road-associated dynamics of orthopteran assemblages with potential implications for ecosystem functioning, we aimed for the following: (1) to test how motorway proximity affects functional diversity of orthopteran assemblages and functional traits related to mobility, feeding guild, lifestyle and moisture preference; and (2) to assess the relationships between trait composition and road-influenced environmental factors. We recorded a significant increase in functional diversity and the occurrence of omnivorous and shrub-dwelling orthopterans, and a decrease in larger ground-dwelling orthopterans at sites close to the motorway. Road-induced changes in vegetation height were identified as the primary driver of these changes. Our findings contribute to a more thorough understanding of the links between road-associated changes in vegetation cover and insect community assembly in grassland habitats.

**Abstract:**

Exhibiting manifold ecological impacts on terrestrial biota, roads have become a major driver of environmental change nowadays. However, many insect groups with high indication potential, such as grasshoppers and crickets (Orthoptera), have been largely neglected in road ecology research from a functional perspective. Using two complementary sampling methods, we have investigated the spatial dynamics of functional diversity and six functional traits in orthopteran assemblages, with respect to motorway proximity and the associated environmental factors, in a grassland habitat in the Lika region, Croatia. This research shows, for the first time, that road proximity can facilitate an increase in the functional diversity of orthopteran assemblages, with shifts in functional traits related to mobility, feeding habits and lifestyle being primarily driven by changes in vegetation height. Our findings also suggest that our ability to detect road-related patterns depends on the choice of a diversity measure and sampling method, since different components of orthopteran assemblages (plant-dwelling vs. ground-dwelling) exhibit different functional responses to road proximity.

## 1. Introduction

The rapid spread of linear transportation networks across the globe has resulted in roads becoming one of the most widespread sources of anthropogenic disturbance in the 21st century [1], representing a global threat to biodiversity [2]. In addition to the more comprehensively studied primary ecological effects of roads on wildlife, e.g., increased mortality, habitat destruction and barrier effects [3,4], there are manifold secondary effects, including chemical, light and noise pollution, alterations in vegetation cover and microclimate, and the spread of exotic species [3,5]. Secondary effects are typically more extensive and arguably more detrimental in the long term, as they change the quality of adjacent habitats within a certain distance from the road, known as a road-effect zone [6,7]. Nevertheless, vegetated strips separating the road from the surrounding landscape (road verges) can also provide new opportunities [8] and constitute important supplementary habitats for grassland arthropods, including pollinators [9,10]. Accordingly, ecological impact assessments based on insect assemblages should contribute to a more precise estimation of the road-effect zone, thus informing roadside management practices [7,11], but are still largely neglected in policy-making and practical conservation alike [12].

In recent decades, functional or trait-based approaches to biotic community assembly have emerged as an important tool in environmental disturbance assessment, offering a novel perspective to community dynamics that are complementary to traditional, species identity-based approaches [13,14]. Trait-based approaches link community and ecosystem functions to environmental change through functional traits, i.e., phenotypic traits that define the interactions between species and their environment, influencing performance or fitness [13,15]. A simplified, function-based community structure thus obtained [16] can be used to quantify functional diversity, as a measure of the value and range of functional traits within a given assemblage [17,18]. This enables comparisons among different ecosystems, providing valuable insights into the patterns and processes affecting assemblages, regardless of their taxonomic composition [17]. Despite the obvious merits of this approach in studying, e.g., the ecology of roadside plant communities [19,20], few studies have attempted to relate road-associated disturbance to functional changes in insect assemblages so far [21].

Grasshoppers and crickets (Insecta: Orthoptera) are frequently used as indicators of anthropogenic disturbance in grassland ecosystems, being highly susceptible to environmental change both at the species and the assemblage levels [22,23]. From the functional perspective, orthopteran assemblages have proved to be indicative of disturbances as disparate as land use change [24,25], flooding [26], fires [27] and urbanisation [28] in various environmental contexts. Likewise, functional responses of orthopteran assemblages to road proximity could provide a good indication of the extent to which such disturbances affect ecosystem processes and their potential for recovery or further degradation [29]. Road ecology research to date, however, has focused mainly on individual orthopteran species used as models in the studies on acoustic interference with traffic noise [30,31], with only a few papers addressing assemblage-level road effects [32].

Previous studies have reported a decrease in Orthoptera biomass with road proximity [7], but also observed that the type of roadside vegetation has a strong influence on orthopteran abundance in urban environments [33]. In a recent study conducted within a grassland ecosystem in the Lika region, Croatia, Rebrina et al. [32] found that a major road can induce changes in the composition and structure of adjacent orthopteran assemblages. Assemblage responses differed depending on both the distance from the road and the assemblage component, i.e., plant-dwelling vs. ground-dwelling; only the latter exhibited significant spatial patterns. In particular, orthopteran abundance, species richness, Shannon diversity and conservation value exhibited a significant decrease at the roadside sites and an increase further away (at 25 m) from the road, likely due to the contrasting effects of vegetation height and traffic noise. Although a trait-based approach would undoubtedly contribute to a better understanding of the species-based patterns described above, to our knowledge it has never been adopted in road ecology research on Orthoptera.

Therefore, expanding on the taxonomically-based approach adopted by Rebrina et al. [32] to study road-associated changes in grassland Orthoptera assemblages sampled by sweep-net and pitfall traps, respectively, we defined two major aims of the current study. Firstly, we aim (1) to investigate the spatial dynamics of functional diversity (evaluated by two distinct diversity measures) and six functional traits (body size, hind femur length to body size ratio, flight capacity, feeding guild, lifestyle and moisture preference) within the same orthopteran assemblages in relation to motorway proximity. We expect to find a disturbance-related increase in functional diversity [34], in line with the intermediate disturbance hypothesis [34,35], and functional traits related to higher mobility and generalist feeding habits [36] within orthopteran assemblages closer to the motorway. Secondly, we aim (2) to relate functional trait composition within orthopteran assemblages to spatial changes in vegetation height, traffic noise and microclimate along the environmental roadside gradient. Our expectation is for the occurrence of smaller shrub-dwelling orthopterans to be positively correlated with higher vegetation and traffic noise near the road, while the occurrence of larger ground-dwelling species is expected to show a negative correlation with these environmental factors [23,37]. By adopting two different sampling methods, we will be able detect potential differences in the functional responses of primarily plant-dwelling vs. ground-dwelling components of orthopteran assemblages.

## 2. Materials and Methods

### 2.1. Study Area and Sampling Design

The field study reported in this paper was conducted in 2018 along the A1 motorway (European route E65) in the Ličko polje karst field (44°30′06.2′′ N 15°32′28.3′′ E), Lika-Senj County, Croatia. The climate of this area is temperate humid with warm summers [38], with the average annual air temperature of 8–9 °C and the average annual amount of precipitation between 1300 and 1400 mm [39]. Orthopterans were sampled at eight locations separated by at least 500 m (to ensure the independence of the samples; see e.g., Hochkirch and Adorf [40]) on the northern side of the motorway (Figure 1), within a mosaic of *Scorzonerion villosae* Horvatić 1949 grasslands and *Arrhenatherion elatioris* Br.-Bl. 1926 meadows [41]. Certain portions of the habitat are currently undergoing succession to heath with *Pteridium aqilinum* (L.) Kuhn (Dennstaedtiaceae Lotsy) [41], due to the abandonment of extensive grazing. When choosing sampling locations, we were careful to select as homogeneous, succession-free areas as possible. There was a higher representation of shrubs and ruderal elements, such as *Rubus* L. spp. (Rosaceae Juss.), *Sambucus ebulus* L. (Viburnaceae Raf.) and *Rumex* L. spp. (Polygonaceae Juss.), closer to the motorway (10–20 m distance) at all locations. At each location, we selected five sites at increasing distance from the motorway (to account for the fact that road proximity effects weaken towards habitat interior, see Knapp et al. [42]), i.e., at 10, 25, 50 and 100 m, including a control site at 500 m distance. Altogether, 40 sites were, thus, selected.

### 2.2. Orthoptera Sampling and Identification

Orthopteran assemblages were sampled using the following two methods: sweep-netting as a standard sampling method for this insect group [43], supplemented by pitfall trapping to obtain a more accurate representation of the ground-dwelling assemblage component [44]. Data obtained by different sampling methods were analysed separately. At each site, sweep-netting was performed in a 30 m linear transect in parallel to the motorway, with a single sweep per meter (altogether 30 sweeps, using a net with 40 cm diameter and 60 cm depth). Three pitfall traps separated by at least five meters were installed along the same transects, consisting of a polyethylene cup (volume 0.4 dm^3^) partially filled with ethylene glycol and water (ratio 3:2), with a drop of detergent to reduce surface tension and a dark roof as protection from rain and litter. In total, 40 sweep-net transects and 120 pitfall traps were used. To cover the main activity period of Orthoptera (June to October) using both methods, we performed the first sweep-net transects and installed pitfall traps at the beginning of June, emptying them once a month until November and simultaneously performing sweep-net transects until October.

The collected Orthoptera material was preserved in 75% ethanol and stored at the Department of Biology, Faculty of Science, University of Zagreb for identification and formal analysis. Orthopterans were identified to the species level using Harz [45,46] and Sardet et al.’s work [47], with the exception of *Chorthippus biguttulus/mollis* (a number of individuals with intermediate characters between the two species), *Poecilimon* sp. (an undescribed species from the *elegans* group) and *Troglophilus* sp. (an immature individual). Early-stage nymphs, rather abundant in the June samples, were assigned to the species they most likely belong to according to their morphology and in line with the following principle: (i) if the nymphs could belong to only one species recorded at the site, they were assigned to this species; (ii) if the site harboured more than one potential species but a single species was dominant in terms of abundance, they were assigned to this species; (iii) if the site harboured more than one potential species with similar abundances, they were distributed evenly among them.

### 2.3. Environmental Measurements

Several biologically relevant, potentially road-influenced environmental factors [48,49] were measured near each pitfall trap (i.e., three measurements per site, averaged to obtain a single value) at each sampling event, which were as follows: (i) air temperature (°C), (ii) air humidity (%), (iii) soil temperature (°C), (iv) soil moisture (%) and (v) average vegetation height (m). Additionally, ambient noise level [50] was recorded at each sampling site for 8–10 min to obtain the following two parameters: (i) average noise level (dBC) and (ii) maximum noise level (dBC; maximum values measured over four intervals of 105 s, averaged to obtain a single value). All measurements were conducted simultaneously with Orthoptera sampling.

### 2.4. Data Analysis

Each of the orthopteran species sampled in this study was assigned a value or a category for the following six functional traits: (i) body size, defined as a geometric mean of body length (mm) across adult males and females of a given species [36], (ii) ratio of the geometric mean of hind femur length (mm) across both sexes to body size, (iii) flight capacity (FC)—full (macropterous), reduced (wing-dimorphic, either due to sexual dimorphism or the occurrence of distinct wing morphs) or none (brachypterous, squamipterous or apterous); (iv) feeding guild—herbivorous, omnivorous or carnivorous; (v) lifestyle—chortobiont (grass-/herb-dwelling), thamnobiont (shrub-dwelling), geobiont (ground-dwelling) or geo-chortobiont (both ground- and grass/herb-dwelling); and (vi) moisture preference—xerophilous, mesophilous or hygrophilous (Table A1 in Appendix A). Body size, femur length and flight capacity determine the mobility of orthopteran species, and thus their capacity to avoid disturbance and colonise novel habitats [51,52]; mobility decreases with an increase in body size and a decrease in femur length and flight capacity [25]. On the other hand, shifts in ecological and life-history traits, such as feeding guild, lifestyle and moisture preference, can indicate changes in resource availability and habitat characteristics as a result of disturbance [26]. Trait assignment was performed in accordance with relevant sources [45,46,53,54,55] and expert knowledge.

Functional diversity of orthopteran assemblages at each sampling site was quantified using the following two standard indices: functional dispersion (FDis) calculated in R 4.0.4 [56] using the FD package [57] and Rao’s quadratic diversity (RaoQ) coefficient [58] calculated in CANOCO ver. 5.11 [52]. Moreover, community weighted mean (CWM) values of each functional trait category per site were calculated in CANOCO ver. 5.11 [59] to assess the shifts in mean trait values within orthopteran assemblages, as a result of environmental selection for certain trait categories [58]. Monthly catches were pooled for each site. The same assemblages were analysed in terms of taxonomic assemblage metrics, with the results published in Rebrina et al. [32].

Data were displayed with standard statistical measures, including mean and standard error, and tested for normality using the Shapiro–Wilk W test. Since none of the environmental factors achieved normal distribution even after transformation (*sqrt*, *4th root*, *log*), differences among the sites were tested using the Kruskal–Wallis H test with multiple comparisons of mean ranks *post hoc* test, as a non-parametric alternative to one-way ANOVA in TIBCO Statistica 13.0 [60].

Generalized linear mixed models (GLMMs) were constructed in SPSS Statistics ver. 27.0 [61] to test for differences in FDis, RaoQ and CWM trait values in orthopteran assemblages among the sites at different distances from the road. These models included sampling locations (level 1) nested within sites (level 2) as subjects, with site (i.e., distance from the road) as a fixed effect and location (i.e., replicate) as a random effect. In case of a statistically significant fixed effect, pairwise contrasts of estimated means among the sites were performed using la east significant difference (LSD) *post hoc* test. GLMMs with normal distribution and identity link function were constructed for target variables ‘FDis’, ‘RaoQ’, ‘body size’, ‘femur/size ratio’, ‘FC reduced’, ‘FC none’, ‘xerophilous’ and ‘mesophilous’, while gamma distribution with log link function was used for the other, non-normally distributed variables. First-order autoregressive (AR1) covariance structure was assumed in all models, since data were collected repeatedly over time (in the course of several months) [62]. Absolute parameter convergence criterion was used for estimation.

In order to assess the influence of environmental factors on the spatial distribution of CWM trait values in orthopteran assemblages, we performed redundancy analysis (RDA) in CANOCO ver. 5.11 [59]. Orthopteran abundances/activity densities were log transformed, centred and standardised by average functional traits. The following factors, identified by Rebrina et al. [32] as both significantly influenced by road proximity and having a significant impact on orthopteran assemblages, were included in the analysis: distance from the motorway, vegetation height, maximum noise level and soil moisture. RDA was followed by variation partitioning to estimate the proportion of variation in functional trait composition explained by individual environmental factors [63], including only the factors with the strongest influence according to the RDA (distance, vegetation height and maximum noise). A Monte Carlo permutation test with 499 permutations was conducted to test for the relationships between trait composition and environmental factors [59].

## 3. Results

### 3.1. Environmental Factors

Changes in environmental factors in relation to motorway proximity were detailed in Rebrina et al. [32]. Accordingly, here we report only the most important findings that are relevant for the scope of the current paper. Soil moisture, vegetation height and noise level (both average and maximum) exhibited significant spatial patterns (Kruskal–Wallis H test with multiple comparisons of mean ranks *post hoc* test, *p* = 0.039 to <0.001). A decrease in soil moisture was observed at 10 m from the road, compared to the more distant sites (50, 100 and 500 m). On the other hand, significantly higher values of vegetation height were recorded at 10 m than at 100 and 500 m distance, whereas noise levels increased gradually with proximity to the road.

### 3.2. Functional Metrics of Orthopteran Assemblages and Motorway Proximity

Functional diversity of orthopteran assemblages quantified by functional dispersion (FDis) exhibited a significant increase with motorway proximity in the sweep-net dataset; Fdis had significantly higher values at 10 m than at 50, 100 and 500 m from the road, and at 25 m than at 500 m distance (Figure 2a, Table 1). We also recorded significant spatial patterns within the following two functional trait categories: feeding guild and lifestyle, in orthopterans sampled by sweep-net (Table 1). Omnivorous orthopterans were significantly more represented in the roadside assemblages at 10 m than at 100 and 500 m distance, and at 25 m than at 500 m from the motorway (Figure 2f, Table 1). A similar pattern was observed for thamnobiont orthopterans, with the sites in closer proximity to the motorway (at 10, 25 and 50 m distance), harbouring significantly more thamnobionts than the control sites (at 500 m; Figure 2g, Table 1). Variance in the random effect was significant in all models from the sweep-net dataset (GLMM, Wald test, Z = 1.997 to −4.491, *p* = 0.046 to <0.001), except for RaoQ (Z = 1.605, *p* = 0.109).

In contrast, we recorded no significant changes in functional diversity (quantified by either FDis or RaoQ) of orthopteran assemblages sampled by pitfall traps related to motorway proximity (Figure 3a,b, Table 2). Body size was the only functional trait that exhibited a significant spatial pattern in the pitfall dataset; CWM values were significantly lower at 25 m than at 100 and 500 m distance, and at 50 m than at 500 m from the motorway (Figure 3c, Table 2). Variance in the random effect was significant in all models from the pitfall dataset (Z = 2.186 to 4.156, *p* = 0.029 to <0.001), except for CWM body size and full flight capacity (Z = 1.764 and 1.888, *p* = 0.078 and 0.059).

### 3.3. Functional Traits and Environmental Factors

In the sweep-net dataset, the first two axes (eigenvalues 0.1731 and 0.0768) of the RDA ordination explained 24.99% of the variation in functional trait composition of the orthopteran assemblages and 88.55% of the variation between functional trait composition and environmental factors. Ordination was statistically significant (pseudo-F = 3.4, *p* = 0.002). Large, herbivorous and chortobiont orthopterans were positively associated with distance from the motorway and soil moisture, and negatively with vegetation height and traffic noise (Figure 4a). The opposite response was observed in omnivorous, thamnobiont orthopterans and those with full flight capacity (Figure 4a).

Likewise, RDA ordination was statistically significant in the pitfall dataset (pseudo-F = 2.7, *p* = 0.004), with its first two axes (eigenvalues 0.1597 and 0.0490) explaining 20.87% of the variation in functional trait composition of orthopteran assemblages and 88.21% of the variation between functional trait composition and environmental factors. Most functional traits exhibited patterns similar to the ones recorded in the sweep-net dataset (see the previous paragraph; Figure 4ab). A positive association with distance from the motorway and soil moisture, and a negative one with vegetation height and traffic noise was recorded in large, flightless and geobiont orthopterans (Figure 4b). Carnivorous, thamnobiont orthopterans and those with long hind femora exhibited a negative response (Figure 4b).

Vegetation height explained the largest portion of the variation in functional trait composition of orthopteran assemblages in both datasets (14.46% in 19.77% total explained variation in the sweep-net and 11.92% in 22.96% total explained variation in the pitfall dataset). It was followed by traffic noise (6.72%) and distance (1.72%) in the sweep-net dataset, and by distance (6.37%) and traffic noise (3.95%) in the pitfall dataset. Statistically significant conditional effects of vegetation height (sweep-net F = 6.6, d.f. = 1, *p* = 0.002; pitfall F = 5.5, d.f. = 1, *p* = 0.001) and traffic noise (F = 3.3, d.f. = 1, *p* = 0.008; F = 3.1, d.f. = 1, *p* = 0.021) were recorded in both datasets, whereas the conditional effect of distance was significant only in the pitfall dataset (F = 4.1, d.f. = 1, *p* = 0.005).

## 4. Discussion

### 4.1. Functional Diversity Patterns with Respect to Road Proximity and Sampling Method

The results of the current study show a significant increase in the functional diversity of orthopteran assemblages closer to the motorway, most likely associated with the increased vegetation heterogeneity at roadside sites [32,64], enabling the co-occurrence of a higher number of functional groups and resulting in a succession-related edge effect [65,66]. This finding is congruous with a similar response observed in ground beetle assemblages sampled within the same study [66], both following the intermediate disturbance hypothesis that moderate disturbance leads to higher species richness [35]. Nevertheless, the investigated roadsides may even be associated with lower disturbance for some taxa, due to the lack of grazing directly adjacent to the road. Interestingly, while we obtained highly consistent results from FDis and RaoQ in ground beetles, the latter exhibited no significant spatial patterns in orthopterans. Although FDis and RaoQ are considered to be robust, highly correlated indices that behave similarly in most cases [67,68], our results suggest they should be used as complementary and not interchangeable tools for measuring functional diversity. Furthermore, functional responses occurred at a smaller spatial scale in orthopterans than in ground beetles (within the first 25 m vs. 50 m from the road) [66], possibly due to the higher mobility of (primarily plant-dwelling) orthopterans, making them less sensitive to the reduction in road-unaffected habitat area [69].

Comparing the results presented here with the results obtained from the same assemblages using taxonomic assemblage metrics, it is evident that orthopterans sampled by sweep-net respond to road proximity mainly in functional terms, and orthopterans sampled by pitfall traps in taxonomic terms (assemblage metrics exhibited a significant decrease at roadside sites; see Rebrina et al. [32]). It seems to suggest the higher functional resistance rather than taxonomic resistance of the ground-dwelling assemblage component, which retains a similar functional structure, despite the changes in taxonomic composition [70]. Alternatively, sweep-netting might simply provide a better indication of functional changes within orthopteran assemblages than pitfall trapping, possibly owing to a more equal representation of species belonging to different lifestyles (except true geobionts) [44]. In line with the results of previous studies on various insect groups in different environmental contexts [36,71], our findings show that a trait-based approach can detect road-associated changes in orthopteran assemblages that are not discernible at the taxonomic level; not only does functional diversity of plant-dwelling orthopterans peak in the roadside environment while their taxonomic diversity exhibits a neutral response [32], but the observed changes in functional trait composition also provide insight into the direction of disturbance-related community assembly [72,73].

### 4.2. Road-Associated Changes in Functional Trait Composition

We recorded a decrease in larger, primarily ground-dwelling orthopterans closer to the motorway, complemented by the negative responses of flightless species and positive responses of species with full flight capacity and longer femora to road proximity, as shown by the RDA. These findings suggest that a higher intensity of road-associated disturbance may act as an environmental filter, selecting against less mobile species and promoting highly mobile ones [36,74]. The association between the mobility-related traits and road proximity, however, was more clearly evident in the pitfall dataset, possibly due to a wider range of Orthoptera size categories efficiently sampled using this method; larger flightless geobionts and geo-chortobionts (e.g., *Arcyptera brevipennis, Gryllus campestris*) were underrepresented in the sweep-net samples (see Rebrina et al. [32]). In particular, this functional group is potentially vulnerable to noise pollution at smaller distances from the motorway [32], due to poorer propagation of acoustic signals at the ground level [75] and/or acoustic interference with traffic noise, arising from the negative correlation between body size and signal frequency [76].

Primarily plant-dwelling orthopterans with omnivorous feeding habits and thamnobiont lifestyles showed a clear preference towards the roadside environment, according to both GLMMs and the RDA. This is probably related to the changes in vegetation height (explaining the largest amount of variation in functional trait composition) and structure, i.e., a transition from typical grassland vegetation away from the road to early-successional roadside vegetation [77]. Major roads can, thus, create a suitable habitat for shrub-dwelling orthopterans (especially ensiferans, such as *Oecanthus pellucens*, *Pachytrachis gracilis*, *P. striolatus* and *Pholidoptera femorata,* in this study) that would not be able to inhabit or would occur in much lower numbers in the surrounding grasslands [65]. Previous studies have suggested the potential of vegetated roadsides and railway verges as both supplementary habitats and refugia for Orthoptera in urban environments [33,78], but our results indicate that this could also be the case in (semi)natural landscapes, at least for certain functional groups. Most of the shrub-dwelling species recorded here are also omnivorous, their generalist feeding habits likely providing an additional advantage in a disturbed environment selecting against specialist herbivores [36]. An exactly opposite response to motorway proximity was recorded in open habitat specialist ground beetles from the same sites [66], further indicating the importance of road-associated changes in vegetation cover for insect community assembly in grassland habitats [72].

## 5. Conclusions

In conclusion, this study provides the first evidence of an increase in functional diversity and corresponding changes in functional trait composition within orthopteran assemblages in response to motorway proximity, with the following two important considerations: (1) the choice of a diversity measure can influence our ability to detect significant patterns (see also Morris et al. [79]); and (2) different components of orthopteran assemblages (primarily plant-dwelling vs. ground-dwelling) show different functional responses to road-associated disturbance. The latter seems to induce functional changes in orthopteran assemblages primarily through modifications in vegetation cover, causing shifts in functional traits related to mobility, feeding habits and lifestyle. Finally, contrasting spatial patterns of both functional and taxonomic assemblage metrics (see [32]) between sweep-net and pitfall datasets suggest that the full extent of road-induced changes in orthopteran assemblages can only be assessed using a combination of sampling methods.

## Figures and Tables

**Figure 1 insects-13-00572-f001:**
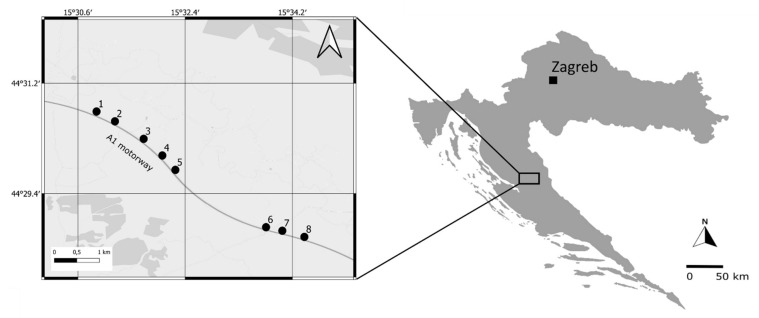
Map of the study area (Ličko polje karst field, Lika-Senj County, Croatia) with the sampling locations marked by black circles.

**Figure 2 insects-13-00572-f002:**
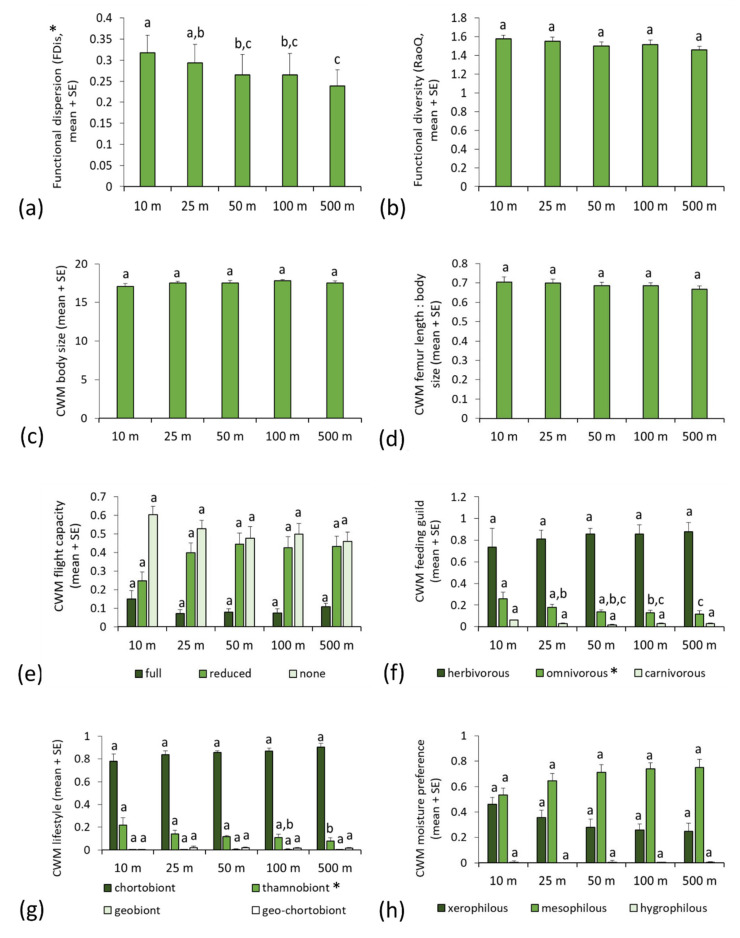
Differences in functional diversity and community weighted mean values of functional traits (mean + standard error) in orthopteran assemblages sampled by sweep-net at different distances from the motorway: (**a**) functional dispersion; (**b**) Rao’s quadratic diversity; (**c**) body size; (**d**) hind femur length/body size ratio; (**e**) flight capacity; (**f**) feeding guild; (**g**) lifestyle; (**h**) moisture preference. Functional trait categories that exhibit significant spatial patterns are marked by asterisks (*). Different letters indicate statistically significant differences among estimated means within each trait category (generalised linear mixed model, least significant difference *post hoc* test, *p* < 0.05).

**Figure 3 insects-13-00572-f003:**
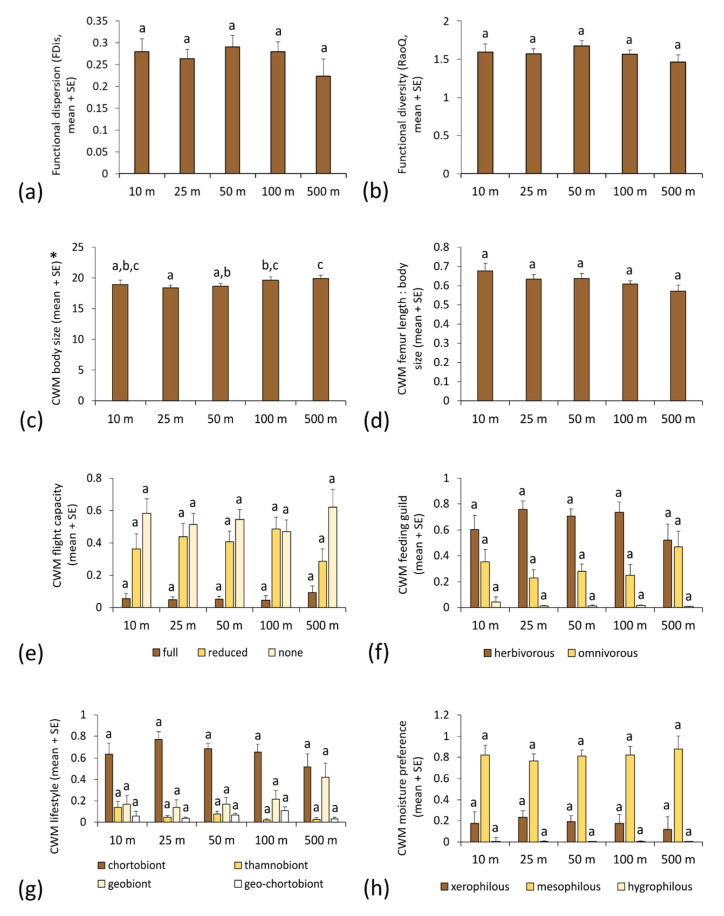
Differences in functional diversity and community weighted mean values of functional traits (mean and standard error) in orthopteran assemblages sampled by pitfall traps at different distances from the motorway: (**a**) functional dispersion; (**b**) Rao’s quadratic diversity; (**c**) body size; (**d**) hind femur length/body size ratio; (**e**) flight capacity; (**f**) feeding guild; (**g**) lifestyle; (**h**) moisture preference. Functional trait categories that exhibit significant spatial patterns are marked by asterisks (*). Different letters indicate statistically significant differences among estimated means within each trait category (generalised linear mixed model, least significant difference *post hoc* test, *p* < 0.05).

**Figure 4 insects-13-00572-f004:**
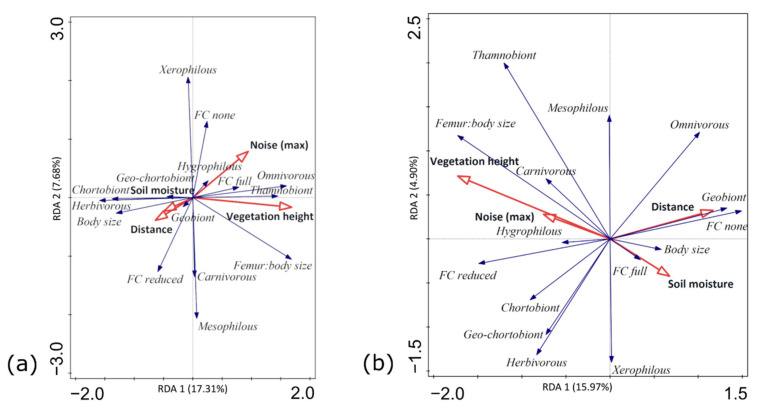
Redundancy analysis (RDA) biplot showing the relationships between functional traits (blue arrows) of orthopterans sampled by (**a**) sweep-net; (**b**) pitfall traps and road-influenced environmental factors (red arrows). Abbreviations: *FC* = flight capacity.

**Table 1 insects-13-00572-t001:** Generalised linear mixed model output showing differences in functional diversity and community weighted mean values of functional traits (with location as a random effect) in orthopteran assemblages sampled by sweep-net at different distances from the motorway (fixed effect). Statistically significant fixed effects and pairwise contrasts obtained from the least significant difference *post hoc* test (*p* < 0.05) are reported in bold. Pairwise contrasts were performed only if the fixed effect was significant. Legend: F—F statistic; d.f.—degrees of freedom; 10, 25, 50, 100, 500—distance from the motorway (m).

Functional Parameter		F	*p*	d.f.	*p*				
					10–25	10–50	10–100	10–500	25–50
Functional dispersion (FDis)		3.189	**0.025**	4	0.310	**0.034**	**0.035**	**0.002**	0.249
Functional diversity (RaoQ)		1.109	0.368	4	/	/	/	/	/
CWM body size		0.805	0.530	4	/	/	/	/	/
CWM femur length: body size		0.594	0.669	4	/	/	/	/	/
CWM flight capacity	full	1.339	0.275	4	/	/	/	/	/
	reduced	2.357	0.073	4	/	/	/	/	/
	none	1.575	0.203	4	/	/	/	/	/
CWM feeding guild	herbivorous	1.750	0.161	4	/	/	/	/	/
	omnivorous	2.790	**0.041**	4	0.451	0.126	**0.041**	**0.004**	0.426
	carnivorous	0.965	0.439	4	/	/	/	/	/
CWM lifestyle	chortobiont	0.412	0.799	4	/	/	/	/	/
	thamnobiont	2.710	**0.046**	4	0.606	0.686	0.238	**0.006**	0.911
	geobiont	0.681	0.610	4	/	/	/	/	/
	geo-chortobiont	1.592	0.198	4	/	/	/	/	/
CWM moisture preference	xerophilous	2.409	0.068	4	/	/	/	/	/
	mesophilous	2.428	0.066	4	/	/	/	/	/
	hygrophilous	1.632	0.188	4	/	/	/	/	/
**Functional Parameter**		**F**	** *p* **	**d.f.**	** *p* **				
					**25–100**	**25–500**	**50–100**	**50–500**	**100–500**
Functional dispersion (FDis)		3.189	**0.025**	4	0.250	0.030	0.998	0.280	0.279
Functional diversity (RaoQ)		1.109	0.368	4	/	/	/	/	/
CWM body size		0.805	0.530	4	/	/	/	/	/
CWM femur length: body size		0.594	0.669	4	/	/	/	/	/
CWM flight capacity	full	1.339	0.275	4	/	/	/	/	/
	reduced	2.357	0.073	4	/	/	/	/	/
	none	1.575	0.203	4	/	/	/	/	/
CWM feeding guild	herbivorous	1.750	0.161	4	/	/	/	/	/
	omnivorous	2.790	**0.041**	4	0.182	**0.028**	0.583	0.148	0.361
	carnivorous	0.965	0.439	4	/	/	/	/	/
CWM lifestyle	chortobiont	0.412	0.799	4	/	/	/	/	/
	thamnobiont	2.710	**0.046**	4	0.501	**0.020**	0.433	**0.016**	0.089
	geobiont	0.681	0.610	4	/	/	/	/	/
	geo-chortobiont	1.592	0.198	4	/	/	/	/	/
CWM moisture preference	xerophilous	2.409	0.068	4	/	/	/	/	/
	mesophilous	2.428	0.066	4	/	/	/	/	/
	hygrophilous	1.632	0.188	4	/	/	/	/	/

**Table 2 insects-13-00572-t002:** Generalised linear mixed model output showing differences in functional diversity and community weighted mean values of functional traits (with location as a random effect) in orthopteran assemblages sampled by pitfall traps at different distances from the motorway (fixed effect). Statistically significant fixed effects and pairwise contrasts obtained from the least significant difference *post hoc* test (*p* < 0.05) are reported in bold. Pairwise contrasts were performed only if the fixed effect was significant. Legend: F—F statistic; d.f.—degrees of freedom; 10, 25, 50, 100, 500—distance from the motorway (m).

Functional Parameter		F	*p*	d.f.	*p*				
					10–25	10–50	10–100	10–500	25–50
Functional dispersion (FDis)		0.871	0.491	4	/	/	/	/	/
Functional diversity (RaoQ)		0.866	0.494	4	/	/	/	/	/
CWM body size		2.908	**0.035**	4	0.208	0.537	0.404	0.084	0.514
CWM femur length: body size		2.048	0.109	4	/	/	/	/	/
CWM flight capacity	full	0.495	0.740	4	/	/	/	/	/
	reduced	0.913	0.467	4	/	/	/	/	/
	none	0.486	0.746	4	/	/	/	/	/
CWM feeding guild	herbivorous	1.584	0.200	4	/	/	/	/	/
	omnivorous	0.725	0.581	4	/	/	/	/	/
	carnivorous	1.131	0.358	4	/	/	/	/	/
CWM lifestyle	chortobiont	2.323	0.076	4	/	/	/	/	/
	thamnobiont	0.974	0.434	4	/	/	/	/	/
	geobiont	0.954	0.445	4	/	/	/	/	/
	geo-chortobiont	2.511	0.059	4	/	/	/	/	/
CWM moisture preference	xerophilous	1.147	0.351	4	/	/	/	/	/
	mesophilous	1.177	0.338	4	/	/	/	/	/
	hygrophilous	1.126	0.360	4	/	/	/	/	/
**Functional Parameter**		**F**	** *p* **	**d.f.**	** *p* **				
					**25–100**	**25–500**	**50–100**	**50–500**	**100–500**
Functional dispersion (FDis)		0.871	0.491	4	/	/	/	/	/
Functional diversity (RaoQ)		0.866	0.494	4	/	/	/	/	/
CWM body size		2.908	**0.035**	4	**0.041**	**0.004**	0.151	**0.022**	0.355
CWM femur length: body size		2.048	0.109	4	/	/	/	/	/
CWM flight capacity	full	0.495	0.740	4	/	/	/	/	/
	reduced	0.913	0.467	4	/	/	/	/	/
	none	0.486	0.746	4	/	/	/	/	/
CWM feeding guild	herbivorous	1.584	0.200	4	/	/	/	/	/
	omnivorous	0.725	0.581	4	/	/	/	/	/
	carnivorous	1.131	0.358	4	/	/	/	/	/
CWM lifestyle	chortobiont	2.323	0.076	4	/	/	/	/	/
	thamnobiont	0.974	0.434	4	/	/	/	/	/
	geobiont	0.954	0.445	4	/	/	/	/	/
	geo-chortobiont	2.511	0.059	4	/	/	/	/	/
CWM moisture preference	xerophilous	1.147	0.351	4	/	/	/	/	/
	mesophilous	1.177	0.338	4	/	/	/	/	/
	hygrophilous	1.126	0.360	4	/	/	/	/	/

## Data Availability

The data presented in this study are available on request from the corresponding author. The data are not publicly available due to the authors’ policy of saving unpublished data for future publications.

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
