# Peer review of "Vegetation Height as the Primary Driver of Functional Changes in Orthopteran Assemblages in a Roadside Habitat"

_insects, 2022, doi:10.3390/insects13070572_

Round 1
Reviewer 1 Report
Dear authors,
I have read this manuscript and I think the theme is quite interesting, however, the way in which it is presented to the journal is not correct. First, in the introductory section you start by writing about the trait-based approach (and never mention anything about functional diversity), then you write about the ecology of roads and the ecological importance of Orthoptera. Immediately, in a simple paragraph of a single sentence, you present a research problem that refers to the fact that they intend to expand a previously published article, which is not a justification or definition of the problem. At the end of the introduction you present the two objectives and the hypotheses, which is fine. However, I do not see any logical sense in the current introduction, I suggest you rewrite an introduction of approximately 6 paragraphs where you clearly define the focus of the research, the model group and the system where the research was carried out.
Regarding materials and methods, I noticed that you refer to the fact that morphometric, morphological and ecological traits can be indicators of ecological functions. However, at no time do they clearly justify it. Ideally, the authors should make a table indicating that the increase or decrease of each trait indicates that in the ecological functions performed by grasshoppers and crickets. Obviously, these functions must be well supported by previous and recent scientific literature and not fall into speculation. This is because, there are many scientific literature about environments shapes morfológicas traits, instead traits affect environment.
Surprisingly, in the discussion and the conclusion of the manuscript, they only write about functional diversity and at no time do they refer to topics seen in the introduction. My recommendation is that the discussion should be closely related to the introduction and should again be approximately 6 paragraphs in length. In short, the introduction must go from the general to the particular, while the discussion from particular to the general, and both must coincide in the way in which the research approach, the model group and the system where the investigation was carried out. In the attached file you will find some minor suggestions.
Sincerely,

Reviewer 2 Report
This paper provides a most useful comparison of orthoptera near roadsides versus fields further away and so should be published with a few revisions as per enclosed comment document.

Round 2
Reviewer 1 Report
line 133. Where was conducted the study? Country and state please. inset area needs a coordinate grid.
You have to work in resolution of figures since they are poor quality. At least 300 dpi in TIFF format. I don't think figures with no significant difference have to in figures. Please remove them from the mosaics.
line 281. please do not write paragraph with a single sentence.
line 228. Did you transform environmental variables? CANOCO works with chi-squared distances, which is very sensitive to non normal data. I think, that heigh of vegetation, and the other environmental factors must be standardized and then normalized for removing the scale effect. Or instead it, use a non parametric unrestricted ordination method.
324. Idem.
